# Predicting Recurrence in Pancreatic Ductal Adenocarcinoma after Radical Surgery Using an AX-Unet Pancreas Segmentation Model and Dynamic Nomogram

**DOI:** 10.3390/bioengineering10070828

**Published:** 2023-07-11

**Authors:** Haixu Ni, Gonghai Zhou, Xinlong Chen, Jing Ren, Minqiang Yang, Yuhong Zhang, Qiyu Zhang, Lei Zhang, Chengsheng Mao, Xun Li

**Affiliations:** 1First Clinical Medical College, Lanzhou University, Lanzhou 730000, China; 2Department of General Surgery, First Hospital of Lanzhou University, Lanzhou 730000, China; 3School of Information Science and Engineering, Lanzhou University, Lanzhou 730000, China; 4The Reproductive Medicine Hospital of the First Hospital of Lanzhou University, Lanzhou 730000, China; 5Department of Preventive Medicine, Feinberg School of Medicine, Northwestern University, Chicago, IL 60611, USA; 6Key Laboratory of Biotherapy and Regenerative Medicine of Gansu Province, Lanzhou 730000, China

**Keywords:** pancreatic ductal adenocarcinoma, pancreas image segmentation, recurrence, pancreatectomy, radiomics

## Abstract

This study aims to investigate the reliability of radiomic features extracted from contrast-enhanced computer tomography (CT) by AX-Unet, a pancreas segmentation model, to analyse the recurrence of pancreatic ductal adenocarcinoma (PDAC) after radical surgery. In this study, we trained an AX-Unet model to extract the radiomic features from preoperative contrast-enhanced CT images on a training set of 205 PDAC patients. Then we evaluated the segmentation ability of AX-Unet and the relationship between radiomic features and clinical characteristics on an independent testing set of 64 patients with clear prognoses. The lasso regression analysis was used to screen for variables of interest affecting patients’ post-operative recurrence, and the Cox proportional risk model regression analysis was used to screen for risk factors and create a nomogram prediction model. The proposed model achieved an accuracy of 85.9% for pancreas segmentation, meeting the requirements of most clinical applications. Radiomic features were found to be significantly correlated with clinical characteristics such as lymph node metastasis, resectability status, and abnormally elevated serum carbohydrate antigen 19-9 (CA 19-9) levels. Specifically, variance and entropy were associated with the recurrence rate (*p* < 0.05). The AUC for the nomogram predicting whether the patient recurred after surgery was 0.92 (95% CI: 0.78–0.99) and the C index was 0.62 (95% CI: 0.48–0.78). The AX-Unet pancreas segmentation model shows promise in analysing recurrence risk factors after radical surgery for PDAC. Additionally, our findings suggest that a dynamic nomogram model based on AX-Unet can provide pancreatic oncologists with more accurate prognostic assessments for their patients.

## 1. Introduction

Pancreatic cancer is the fourth leading cause of cancer-related mortality in the world, and the five-year survival rate is the lowest among all malignant tumours, which is 11% [1]. The most common type of pancreatic exocrine tumour is pancreatic ductal adenocarcinoma (PDAC), and its five-year recurrence rate after radical surgery is 85% [2]. Identified risk factors associated with recurrence after radical surgery known to increase the risk of recurrence after radical surgery include tumour size, lymph node metastasis, vascular invasion, and abnormally elevated serum tumour markers [3,4]. Previous studies have shown that diabetes mellitus (DM) or hyperglycaemia is relevant to the pathogenesis of PDAC. However, there is still a debatable topic on whether DM has an impact on the recurrence after radical surgery [5]. Previous research has ascertained a strong correlation between resectability status and the prognosis after radical surgery [6]. Therefore, further research is required to establish the relationship between resectability status, prognosis, and the appropriate treatment.

Artificial intelligence (AI) technology has been adopted in many areas of modern society, especially in image processing [7,8]. Contrast-enhanced CT scans are the most commonly used imaging modality for PDAC diagnosis and resectability evaluation. Therefore, AI can be used to process preoperative CT data. For example, Liu et al. [9] used multi-centre pancreas CT images to train a convolutional neural networks (CNN) model to screen small tumours from pancreatic tissue, achieving a sensitivity of 92.1%, which was much higher than human recognition, thus helping surgeons to avoid residual small tumours at the resection margin.

Radiomics are capable of extracting high flux features and quantitatively analysing region of interests (ROIs) to accurately reflect tumour heterogeneity and biological characteristics [10,11]. However, manual ROI selection is still prevalent in many pancreatic tumour radiomics studies, making researchers’ work more demanding and subject to biases due to the high degree of subjectivity in the process [9,12,13]. Image segmentation is an effective solution to these challenges, enabling radiomics to eliminate interference from other areas and enhance the precision of clinical models designed for pancreatic diseases [14]. As the use of radiomics expands, it is becoming more common to study image features and their correlations to clinical features in clinical studies. The utilization of artificial intelligence (AI) alongside radiomics presents an opportunity to uncover new patterns within medical images, ultimately enhancing the role of radiology in treating cancer patients [15]. Therefore, AI could potentially be a new trend in the development of radiomics by assisting in ROI selection.

In the realm of pancreatic segmentation, numerous approaches have been proposed. Farag et al. [16] utilized a dropout-enabled CNN model to classify pixels at a granular level. Cai et al. [17] introduced a convolutional LSTM network integrated into the CNN’s output layer, enabling segmentation of 2D slices of the pancreas. However, these methods often neglect the spatial information across slices as they solely rely on merging information from 2D CT image slices. Man et al. [18] put forth a CNN-based coarse-to-fine classifier designed for analysing image patches and regions. Zhang et al. [19] proposed an efficient SegNet network composed of a basic encoder, slim decoder, and an efficient context block. Although these methods partially integrate spatial information, there is still room for improvement in making accurate boundary segmentation decisions. Shi et al. [20] introduced CoraNet, a semi-supervised segmentation model that leverages uncertainty estimation and a separate self-training strategy, without relying on predefined boundary-aware assumptions. In contrast to prior methods, our framework excels in extracting comprehensive spatial and channel features, implementing multi-level and multi-scale feature extraction. Our approach has yielded outstanding results on both our proprietary dataset and publicly available datasets.

Unet is an FCN variant that displays good performance in medical image processing [21]. In comparison to other neural network algorithms, Unet has several advantages. Firstly, the acquisition and processing of medical image features are relatively difficult compared to traditional manual labelling tasks, such as object detection and segmenting organs, requiring a certain level of expertise. Difficulties such as these make it challenging to create large-scale image annotation datasets [22]. Therefore, the Unet model structure is relatively simple and more appropriate for situations involving less data. The model parameters must be minimized to avoid overfitting. Secondly, medical image semantics that lack complex backgrounds but contain multiple fixed organ structures are relatively simple and stable in their composition. Due to the limited semantic information available, high-level semantic information and low-level features are both important (the skip connection and U-shaped structure of Unet are useful) [23]; (3) Finally, interpretability is an essential factor in clinical research model construction.

In the domain of pancreatic segmentation, a multitude of methodologies have been proposed. Farag et al. [16] employed a convolutional neural network (CNN) model with dropout to achieve precise pixel-level classification. Cai et al. [17] introduced a convolutional long short-term memory (LSTM) network integrated into the CNN’s output layer, enabling the segmentation of two-dimensional (2D) slices of the pancreas. However, these methods often overlook the spatial information across slices as they solely rely on merging information from 2D CT image slices. Oktay et al. [24] introduced a groundbreaking attention gate (AG) model specifically designed for medical imaging. This model autonomously learns to concentrate on structures with varying shapes and sizes. Notably, it achieves remarkable enhancements in the performance of Unet, rendering it of paramount importance for PDAC image segmentation. Man et al. [18] proposed a CNN-based coarse-to-fine classifier specifically designed to analyse image patches and regions. Zhang et al. [19] proposed an efficient SegNet network composed of a basic encoder, slim decoder, and contextually efficient block. Although these methods partially incorporate spatial information, there is still scope for enhancing boundary segmentation decisions. Ribalta Lorenzo et al. [25] presented a two-step multi-modal Unet-based architecture with unsupervised pre-training and a surface loss component, effectively leveraging all magnetic resonance modalities for brain tumour segmentation. Shi et al. [20] introduced CoraNet, a semi-supervised segmentation model that utilizes uncertainty estimation and a separate self-training strategy without relying on predefined boundary-aware assumptions. Isensee et al. [26] pioneered a self-configuring segmentation approach that encompasses preprocessing, network architecture, training, and post-processing to adapt to diverse tasks. This method has exhibited exceptional performance on various publicly accessible biomedical segmentation datasets.

We have developed a new deep learning framework known as AX-Unet for pancreas CT image segmentation, integrating the strengths of deepLabV series [27], Unet, and Xception networks [28]. This approach aids physicians in detecting pancreatic tumours and evaluating the correlation between image features and clinical characteristics. We further designed a nomogram prediction model with risk factors, enabling us to determine the likelihood of patient recurrence. The objective of our research is to evaluate the significance of CT images in post-operative recurrence and their impact on patient outcomes.

## 2. Materials and Methods

### 2.1. Ethical Approval

Institutional review board (IRB) approval was obtained before the collection of the dataset (LDYYLL2021-471). The IRB of the first hospital of Lanzhou University approved this study and waived the need for informed consent.

### 2.2. Patient Participation

#### 2.2.1. AX-Unet Model Training Patient Data Source Patient Description

We included patients with pancreatic tumours treated at the Department of Hepatobiliary and Pancreatic Surgery, First Hospital of Lanzhou University between November 2017 and November 2020 for data collection and processing. Inclusion criteria were: (1) complete contrast-enhanced CT scan data, (2) diagnosis of PDAC by preoperative endoscopic ultrasonography-guided fine-needle aspiration (EUS-FNA) or post-operative pathology, (3) no distant metastasis, and (4) no preoperative invasive treatments. Patients who did not undergo standardized pancreas protocol multi-slice computed tomography (MDCT) or were definitively diagnosed with distant metastases were excluded from the study to ensure cohort homogeneity. In total, 205 patients were selected to train of the AX-Unet segmentation model, with 80% of the patients randomly assigned to the training group and 20% to the validation group.

#### 2.2.2. Patient Recurrence Analysis

To explore the effectiveness of the AX-Unet pancreas segmentation model in the recurrence of PDAC after radical surgery, we screened patients who underwent radical surgery from December 2021 to December 2022 in the hospital. Inclusion criteria: (1) after pathological diagnosis with PDAC, the status of resectability was conducted by experienced radiologists during multidisciplinary team (MDT) discussion, and chief surgeons evaluated the difficulty of the operation and finally performed radical surgery; (2) complete medical records, image data, therapeutic process, follow-up information, and definite recurrence date. Follow-up criteria: The patients are followed-up according to the routine follow-up procedures of our department after surgery. The follow-up items included tumour-related serological markers and imagological examinations (contrast-enhanced CT scans of the upper abdomen and chest). The patient’s age, gender, body mass index (BMI), serum CA19-9 level, and status of resectability were recorded. Detection of tumour markers: A radioimmunoassay was used to detect the level of CA19-9 in all patients one week before surgery. The upper limit of the normal value of CA19-9 was defined as 37 U/mL [29]. The status of resectability was in reference to the 2021 NCCN guidelines, with all patients graded according to the tumour invasion of main vessels during the MDT discussion. Exclusion criteria: incomplete medical records, follow-up loss, or unclear recurrence time. The clinical principles and guidelines for the treatment of pancreatic cancer suggested that surgery was no longer recommended when distant metastases were present, instead individualized pharmacological treatment wa carried out. This is similar when lymph node metastases were present, and surgery was not recommended when lymph nodes in distant organs were enlarged and accompanied by organ tumour metastases. Local lymph node metastases can be treated by resection of the primary site and by extended debulking.

### 2.3. Establishment of the AX-Unet Pancreas Segmentation Model Based on Sequential Contrast-Enhanced CT Scans for Patients Diagnosed with PDAC

The AX-Unet is a novel deep learning framework that focuses on accurately segmenting pancreas CT images to assist physicians in screening for pancreatic tumours. As shown in Figure 1C, the framework combines deepLabV series, Unet, and Xception networks and preserves Unet’s encoder–decoder structure while introducing a modified atrous spatial pyramid pooling (ASPP) [27] module to deal with downsampling issues and extract multi-level contextual information. To achieve full feature extraction and decouple channel information, a special group convolution [30,31] operation is used. To overcome the issue of indistinct boundaries, an explicit boundary-aware loss function is employed. In medical imaging, the extraction and analysis of features are crucial for lesion detection and diagnosis. However, factors such as scale and background information can affect feature recognition. To address this issue, several methods have been employed, including scale normalization, scale-space analysis, background subtraction, and contextual information guidance. Scale normalization ensures consistent size and resolution, while scale-space analysis captures features at different scales. Background subtraction helps eliminate noise and interference, while contextual information provides valuable insights. These techniques improve the accuracy of feature recognition and contribute to precise lesion detection and diagnosis. The proposed model outperformed state-of-the-art methods in two public datasets [32]. This study provides a significant contribution to the medical image analysis field and has the potential to aid physicians in the early detection and diagnosis of pancreatic tumours. Thus, using the AX-Unet framework to diagnose PDAC patients could be useful.

We trained the model using patient data obtained from the source described in Section 2.2.1. Reference standard segmentations were manual slice-by-slice tracings of the pancreas and peri-pancreatic vessels by two radiologists with 10 years of experience using ImageJ (Version 1.53n 7), they were blind to the patient information in the segmentation process. The entire pancreatic parenchyma was considered as a ROI during the pancreatic phase of the CT protocol, avoiding common bile duct, visible blood vessels, and fat space around the pancreas. These delineations have been taken as the reference standard in all tests.

For each patient in the training group, we identified and extracted the CT images of the pancreas in the enhanced scans and portal venous scans, and excluded images that did not meet the minimum requirements. The image processing included: (1) selecting the ROI area of the CT images; (2) obtaining pancreatic image features by the AX-Unet model. The features include (1) textural features: grey level co-occurrence texture matrix (GLCM), grey level run length matrix (GLRLM), grey level size zone matrix (GLSZM), and grey level dependence matrix (GLDM); (2) histogram features: HU value, mean, energy, entropy, variance, etc.; (3) geometric shape: sharpness, superficial area, etc., (Figure 1A).

### 2.4. Assessment of the Effectiveness of the AX-Unet Segmentation Model

The extraction of image feature heterogeneity in tumour recurrence and prognosis aids in distinguishing between benign and malignant lesions, visualizing recurrence and metastasis sites, predicting treatment response, and prognosticating patient outcomes. Overall, it contributes to more accurate diagnoses, treatment planning, and prognosis prediction in the field of medical imaging.

In this study, we conducted an evaluation of our proposed approach for pancreas segmentation by employing various metrics that are widely used in medical image segmentation research. We used the Dice similarity coefficient (DSC) as the primary metric to measure the similarity between the predicted and ground truth segmentations. Our evaluation of the approach included reporting the average, maximum, and minimum DSC scores across all testing cases in the dataset to provide a comprehensive analysis. In particular, the DSC score was computed by dividing the intersection of the predicted segmentation (Z) and ground truth segmentation (Y) by the average of the total number of voxels in both segmentations.
$$\mathrm{DSC}\left(Z,Y\right)=\frac{2\times |Z\cap Y|}{\left|Z|+|Y\right|}$$

Besides DSC, we employed the Jaccard coefficient, precision, and recall as additional metrics to further evaluate the performance of our approach. The Jaccard coefficient is a metric utilized to ascertain the similarity between the real and predicted pancreatic areas at the pixel level. The Jaccard coefficient is calculated by dividing the intersection of the real and predicted pancreatic areas by their union. We used U to represent the real pancreatic area and V for the predicted pancreatic area. Additionally, precision and recall are widely used to evaluate the performance of binary classifiers, and we used them in the study to assess the accuracy of our approach in identifying true positive (TP), false positive (FP), and false negative (FN) instances.
$$\begin{array}{c} \;\mathrm{Jaccard}\;\left(U,\;V\right)=\frac{\left|U\cap V\right|}{\left|U\cup V\right|}\\ \;\mathrm{Precision}\;=\frac{TP}{TP+FP}\\ \;\mathrm{Recall}\;=\frac{TP}{TP+FN} \end{array}$$

In our previous study, we reported the DSC, Jaccard, recall, and precision values of the AX-Unet model on the NIH and MSD datasets separately, and demonstrated its significant superiority in these metrics compared to other models [32]. The NIH pancreas segmentation dataset from TCIA is an open-source dataset widely used for comparing methods in pancreas segmentation [33]. It involves 82 CT volumes with a spatial resolution of 512 × 512 × L and slice thickness ranging from 0.5 to 1.0 mm. The dataset uses a standardized four-fold cross-validation with three folds for training and one for testing. Additionally, the Medical Segmentation Decathlon (MSD) challenge evaluates machine learning algorithms for ten different semantic segmentation tasks, including the pancreas part in portal venous phase CT from the Memorial Sloan Kettering Cancer Center [34]. The official training–test splits for the MSD include 281 subjects in the training set and 139 subjects in the test set. Our study’s use of these established evaluation metrics strengthens its rigour and scientific validity. Since our previous work was to acquire local images of the pancreas from the enhanced CT images of the pancreas by AX-Unet, this work was performed based on the database (LDYYLL2021-471), and the segmented images of the pancreas in this database do not contain the pancreatic vessels, so the model constructed based on this database will produce the same results for the segmentation of the image data from our hospital.

We conducted recurrence analysis on the patients mentioned in Section 2.2.2. Specifically, a total of 64 patients were included in the recurrence analysis, and these patients were distinct from those used in the previous AX-Unet training model. The main focus of the analysis was to assess the mean recurrence-free survival (mRFS) as the primary outcome (Figure 1B).

### 2.5. Statistical Methods

The image features and clinical data were analysed using SPSS 22.0 software. The measurement data were presented as mean ± standard deviation (x ± s). If the data exhibited normal distribution, the independent sample *t*-test was used. Conversely, the Mann–Whitney U test and the Kruskal–Wallis test were used when the distribution was non-normal, presented with the median (minimum, maximum). Categorical data were analysed using the chi-squared test. The Pearson linear correlation analysis method and lasso regression were utilized to evaluate the correlation between the image features. If the coefficient was ≥0.8, the correlated feature was selected randomly and excluded. Kaplan–Meier curves was used to perform univariate survival analysis and the log-rank method was used for hypothesis testing in comparing two groups. Any significant variable in the univariate analysis was a potential candidate for multivariate analysis. The Cox proportional risk model was used for multi-factor prognostic analysis. The DynNom package implemented a web-based version of the nomogram for this analysis. We used the rms package to build a scoring nomogram model. The c-index and the area under the receiver operating characteristic curve (AUC) were common measures. Consistency was visualized in the form of calibration curves. The nomogram can be interpreted as follows: (1) a vertical line is drawn on the score axis, (2) a score is assigned to each predictor, (3) the total score is summed, and (4) a vertical line is drawn from the axis of the total score corresponding to the value at the lower end of the column plot. This value is the probability of post-operative recurrence. The maximum value of the c-index is 1.0, and the higher the c-index, the better the model’s predictive ability. We considered all tests with a *p* value < 0.05 as statistically significant.

## 3. Results

### 3.1. The Effectiveness of the Segmentation Model

The performance of the AX-Unet network architecture for the specific scenario of pancreas segmentation was improved by our proposed optimization. The high- and low-level features of organ segmentation are both important in clinical problems [35]. Therefore, we added a parallel feature extraction module and residual structure to realize the parallel extraction and fusion of high- and low-level features. Specifically, we refer to the parallel feature extraction ASPP module [36], which extracts features in parallel through dilated convolution with different ratios when the feature map size is down to 64 in the coding process of AX-Unet. In the process of downsampling and upsampling, the residual structure is added to realize the fusion of different levels of features through concatenation [37]. In addition, this study added an attention mechanism in the process of parallel feature extraction to distinguish the importance of different levels of information in the model, applying learnable weights to the feature maps from different levels. Finally, the Dice score reached 85.9% (Table 1) for our dataset, meaning it can be applied to clinical research and work (Figure 2).

The results depicted in Table 1 provide compelling evidence that the AX-Unet model consistently outperforms Unet, Bottom-up, Attention Unet, and nn-Unet across multiple evaluation metrics, including DSC, Jaccard, recall, and precision. Notably, the AX-Unet model exhibits a remarkable 20% improvement in DSC compared to the Unet model, underscoring its significant advantage. Furthermore, even when compared to the widely acclaimed nn-Unet model, recognized for its exceptional performance in the field of image segmentation, the AX-Unet model demonstrates a noteworthy 6% enhancement in DSC. These findings undoubtedly highlight the exceptional effectiveness and proficiency of the AX-Unet model specifically in the context of PDAC image segmentation.

### 3.2. Model Training Details

Our model was trained on the PaddlePaddle platform with multiple cards, using four Tesla V100 GPUs. We used the whitelist provided by PaddlePaddle to achieve mixed-precision training acceleration. Source code: https://github.com/zhangyuhong02/AX-Unet.git accessed on 31 May 2023.

In addition, we used WGIF [38] to enhance the picture and obtain an image with stronger texture features to facilitate training after completion. Figure 3 and Figure 4 illustrate the comparison between the original and enhanced images, showcasing the augmentation of strong classification texture features. The original image, as depicted in Figure 3, demonstrates limited texture details and a relatively lower discriminative potential. However, upon applying advanced image enhancement techniques, as depicted in Figure 4, the texture features undergo substantial enrichment, resulting in heightened discriminative capabilities.

### 3.3. General Clinical Features of the Subjects

Among the 64 PDAC patients included in this study, 41 were males and 23 were females; the age was 56.5±9.08 years; the average BMI was $22.74\pm 2.97\;\mathrm{Kg}/m^{2},20$ had $\mathrm{BMI}>24\;\mathrm{Kg}/m^{2}$, and 44 had a BMI $\leq 24\;\mathrm{Kg}/m^{2};31$ cases had a history of DM; 33 cases had no DM; tumours were localized to the head of the pancreas in 58 cases and to the body or tail in 6 cases; 20 cases had a baseline CA19-9 ≤37U/mL, while 44 cases were >37U/mL. It was found that 33 patients had with resectable pancreatic cancer (RPC) and 31 patients had borderline resectable pancreatic cancer (BRPC) in the MDT data collection; post-operative pathological results showed that there were 7 patients with high differentiation, 21 patients with moderate differentiation, 36 patients with poor differentiation, 30 patients with lymph node metastasis and 34 cases without lymph node metastasis. Most of the patients (58 cases, 90.6%) underwent pancreaticoduodenectomy. The follow-up date ended on 31 December 2021. At the last follow-up, 46 of the 64 patients recurred, the median follow-up time of the entire cohort was 14.5 months, and there were 4 cases (6.3%),16 cases (25%), and 10 cases (15.6%) of recurrence 3, 6, and 9 months, respectively. The results are shown in Table 2.

### 3.4. Analysis of Image Features Extracted from the AX-Unet Model

Table 3 illustrates the inclusion of the Hu value, contrast [39], entropy, and variance after screening image features through Pearson and lasso regression analysis of the AX-Unet pancreas segmentation model. Statistically significant differences in contrast were found across different genders and levels of obesity (*p* < 0.05). The study found that both the mean value and variance of DM patients were significantly distinct from non-DM patients (*p* < 0.05). Moreover, patients with elevated CA19-9 showed significant increases in their Hu, entropy, and variance values (*p* < 0.05). Furthermore, differences in the Hu value were shown to be significant among patients with varying pathological grades (*p* < 0.05). Entropy was also found to be relevant to whether patients had lymph node metastasis (*p* < 0.05). Lastly, when using recurrence as the grouping basis, the variance and entropy of pancreas images taken before surgery manifested as significant associations with recurrence (*p* < 0.05).

### 3.5. Univariate Analysis

Table 2 presents the result of the study. Univariate analysis of the clinical characteristics conducted throughout the study revealed a significant association of resectability, CA19-9 levels, and history of DM with the prognosis. However, age, BMI, intraoperative bleeding, and operation time, as well as other factors, were not significantly correlated with recurrence. Additionally, the study found that DM patients had a markedly poorer mRFS than non-DM patients. (24.84 months vs. 11.61 months, $\chi^{2}=7.32,p$ = 0.007), a difference was also found when comparing the two groups with the 3-,6- and 9-month recurrence rates (3.0%vs.9.6%,22.5% vs. 27.3%,12.1% vs. 19.3%,p < 0.05). When patients were grouped by resectability status, the level of serum CA19-9 and the rate of lymph node metastasis in BRPC patients were significantly higher than in RPC patients, and their mRFS also exhibited significant differences when compared with RPC patients, 27.22 months and 10.71 months, respectively, $\left(\chi^{2}=14.801,p<0.05\right)$. Furthermore, the status of BRPC was associated with early recurrence (p<0.05). When compared with normal levels, patients with elevated preoperative serum CA19-9 had a higher lymph node metastasis rate and an early recurrence rate (p<0.05),mRFS was also shorter (37.20 months vs. 10.39 months, $\left.\chi^{2}=24.467,p=0.01\right)$. At the same time, the recurrence rate of patients with lymph node metastasis was significantly higher than that of non-lymph node metastasis, and mRFS was also different between the two groups $\left(\chi^{2}=11.354,p<0.05\right)$ (Figure 5).

### 3.6. Multivariate Analysis

Risk factors (DM, resectability status, CA19-9 level, pathological type, lymph node metastasis) in the univariate analysis were included in a multivariate analysis. CA19-9 > normal baseline was an independent risk factor for tumour recurrence after PDAC radical surgery (HR = 5.13 (2.05–12.84), *p* = 0.01) (Table 2).

### 3.7. Logistic Regression Analysis

Analysing the recurrence of 64 PDAC patients after radical surgery, male patients had a higher risk of recurrence than female patients (OR = 9.45, 95% CI: 1.01–88.71), and BRPC patients had a significantly higher risk of recurrence than RPC patients (OR = 19.88, 95% CI: 1.52–260.35). Patients with increased entropy had an increased risk of recurrence (OR = 0.01, 95% Cl: 0.002–0.03). The Hosmer–Lemeshow test found that current data were fully extracted (*p* < 0.05) (Table 4).

### 3.8. Development and Evaluation of a Post-Operative Recurrence Scoring System

In this study, Cox proportional hazards regression analysis was performed to identify and obtain the top three risk factors associated with post-operative recurrence following radical surgery. Using the rms package in R, a nomogram was created along with a web-based scoring tool called DynNom for online use. https://nomogrampancreaticcancer.shinyapps.io/DynNomapp accessed on 31 May 2023, is freely accessible to the public for scoring purposes. The dynamic nomogram enables the assessment of an individual’s probability of post-operative recurrence and the reduction of such probability through active treatment of high-risk patients with adjuvant therapy. The model’s discrimination was evaluated using AUC: 0.84 and 95% CI: 0.78–0.99. This indicates a strong predictive capability of the nomogram model for post-operative recurrence. In addition, the bootstrap internal validation method was employed to validate the scoring system, and the resulting c-index was 0.62 with a 95% CI of 0.48–0.78.

## 4. Discussion

At present, most radiomic studies use the largest lesion layer as the ROI to perform feature extraction [40,41]. Therefore, it cannot represent the overall characteristics of the entire organ or lesion. The current ROI selections are mainly conducted manually, semi-automatically, and automatically, but most radiomic studies on the pancreas are based on manual selection because there are many surrounding interfering factors, and the shape varies greatly among individuals.

Disparate from prior approaches, AX-Unet surpassed them by extracting more comprehensive spatial and channel features. This capability allowed the model to capture a richer understanding of the image content, encompassing intricate details and contextual information crucial for accurate segmentation. Moreover, AX-Unet introduced a multi-level and multi-scale feature extraction scheme, enabling it to effectively handle objects of varying sizes within images. By considering both local and global contexts, the model comprehended spatial relationships and captured hierarchical structures, resulting in more precise and coherent segmentation outcomes. These advancements in feature extraction and boundary evaluation have proven to be highly effective, as demonstrated by the excellent results achieved by AX-Unet on multiple public datasets as well as our own dataset. Overall, AX-Unet stood out as a state-of-the-art solution, outperforming previous methods and demonstrating its efficacy in the field of PDAC image segmentation.

The pancreas was segmented using the AX-Unet model, and radiomic features were extracted via OpenCV-Python, utilizing upper abdominal contrast-enhanced CT scans. Our model incorporates an ASPP module designed to extract features in parallel at multiple levels while maintaining high-resolution feature maps. By implementing depth-wise separable convolution, the model is capable of fully decoupling information between channels, thereby enhancing the network’s performance, as confirmed by our experimental evaluation. Through this approach, the AX-Unet segmentation model can effectively address complex tasks, reduce workload, improve delineation accuracy, and minimize the potential for subjective errors.

By employing the AX-Unet model, physicians can accurately identify and segment tumour regions, enabling personalized treatment plans for patients. Utilizing the AX-Unet model for image segmentation allows physicians to precisely determine the location, shape, and size of tumours. This aids in devising surgical plans with enhanced accuracy, minimizing damage to surrounding healthy tissues while maximizing tumour removal. Additionally, AX-Unet assists in defining lesion boundaries for radiation therapy, thereby improving the precision and effectiveness of treatment. The tumour segmentation images generated by the AX-Unet model can also be employed to develop personalized treatment strategies. By analysing tumour characteristics and distribution, physicians can tailor treatment approaches to individual patients, considering their specific circumstances. This includes selecting the most suitable chemotherapy drugs, dosage, and treatment regimens to optimize therapeutic outcomes while reducing side effects. Furthermore, AX-Unet-based image segmentation contributes to the development of more precise medications. By scrutinizing tumour segmentation images, researchers can gain deeper insights into the biological features and pathological processes of tumours, uncovering novel treatment targets and drug receptors. This establishes the foundation for personalized drug development, enabling highly targeted and individualized therapies that significantly enhance the prognosis of cancer patients.

The study revealed that patients with DM had statistically significant differences in the mRFS at 3, 6, and 9 months of recurrence (*p* < 0.05). The increased drug resistance and migration ability of tumour cells caused by elevated blood glucose may be the attributed cause for these differences. Analysis of image features showed that there was a significant correlation (*p* < 0.05) between the Hu value and variance of the pancreatic CT images and the patient’s history of DM. These findings are relevant to the degree of fibrosis and lipid deposition in the whole pancreas [42,43].

The CA19-9 level is an important serological indicator for PDAC diagnosis and prognosis. Its standard level is <37 U/mL [44]. Abnormally increases in serum CA19-9 was related to lymph node metastasis and had a significant effect on mRFS (*p* < 0.05) in the univariate analysis. There is still a statistical significance after excluding the interference of other risk factors in the multivariate analysis (*p* < 0.05). Therefore, a higher CA19-9 level can be used as a prognostic factor for patients after radical surgery. CT image features such as Hu value, entropy value, and variance can also indicate abnormally increases in CA19-9 (*p* < 0.05). Therefore, image features may be used to assess the prognosis of PDAC patients. Considering that the Lewis antigen and CA19-9 levels were not significantly elevated in some PDAC patients [45], further research is needed to explore more image features to assess such PDAC patients’ prognosis.

The resectability status of a patient has a significant impact on their prognosis after radical surgery. In the univariate analysis, the prognosis for BRPC patients was markedly lower than that of RPC patients (*p* < 0.05), and this was found to be accompanied by a higher incidence of early recurrence (*p* < 0.05). An explanation for this may be the proximity of the BRPC tumour to major blood vessels which increases its likelihood of early recurrence. Hence, accurately staging RPC and BRPC patients is a crucial aspect of future studies to enable appropriate treatment decisions for BRPC patients. Additionally, it was discovered that the entropy value of pancreas CT images before surgery was linked with different resectability statuses and may serve as a supplementary index for precise evaluation in the future [46].

PDAC can spread through the lymph nodes and surrounding nerves, in addition to blood metastases. Extended radical lymphadenectomy has been reported to improve patient survival rates [47]. In this study, the univariate analysis showed that the mRFS of lymph-node-metastasized patients was significantly lower than that of patients with no lymph node metastasis (*p* < 0.05). In the analysis of image features, entropy was found to have a significant relationship with lymph node metastasis (*p* < 0.05), presumably because pancreatic texture changes when lymph node metastasis is present.

Pathological grading currently holds a crucial role in predicting PDAC prognosis. Prior radiomic studies have revealed that tumour margin sharpness is associated with the degree of pathological differentiation [48]; however, no research has indicated whether overall pancreas margin sharpness is related to pathological grade. The current study identified a correlation between the post-operative pathological grade and the Hu value of CT images of the entire pancreas (*p* < 0.05). Thus, findings from this study can be introduced to formulate preoperative prediction models based on CT images and, in turn, improve the predictive accuracy of pathological grades before surgery.

Radiomic features have a certain efficacy for the prognostic analysis of various types of malignant tumours [49,50,51]. This study found that the image features extracted from the AX-Unet segmentation model have a certain significance for evaluating the recurrence of PDAC after radical surgery. As shown in other studies that the entropy of images can reflect textual irregularity and link to heterogeneity issues, it may also be associated with the recurrence of colorectal cancer, gynaecological cancer, and endometrial cancer after surgery [52,53]; therefore, the features and parameters of tumour texture have proven that the prediction of prognosis before surgery can be achieve by the combination of image features and clinical data, potentially reducing the difficulty for clinicians in patient follow-ups and optimize the management of patients with PDAC.

Recently, nomogram models have gained widespread use in tumour prediction modelling due to their powerful simplification and excellent predictive power [54]. In this study, the nomogram had a c-index of 0.62, indicating an average predictive power, while the higher AUC value denotes a better predictive accuracy for the model. To attain a more precise assessment of patient prognosis in clinical practice and to direct individualized treatment, nomogram prediction models should incorporate additional risk factors.

This study has certain limitations: (1) First, as a retrospective study, the treatment decisions of the MDT vary with personal cognition and guideline updates; (2) at present, the median survival time (mOS) of patients with R0 resection is 22–23 months, but this was not included in this study, so longer follow-ups are needed for future studies.

## 5. Conclusions

The aim of this study was to identify the value of an AX-Unet pancreas segmentation model applied in preoperative CT on predictions of tumour recurrence after resection of PDAC, which may be a new intersection of radiomics and AI in the future. Combining the advantages of the two disciplines and designing a strong interpretability algorithm could avoid subjective interference in the research process and reduce the work intensity of clinical practitioners. By making full use of image features and clinical characteristics to analyse the prognosis of patients, the upcoming models could assist physicians in regular post-operative prognosis, and treat high-risk patients as soon as possible, thus improving the overall prognosis of PDAC patients.

## Figures and Tables

**Figure 1 bioengineering-10-00828-f001:**
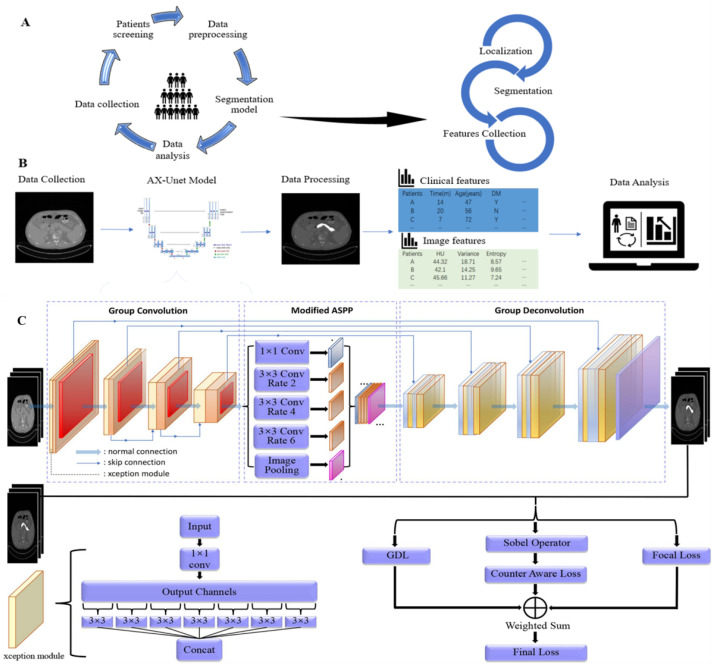
Procession of data processing and analysis. (**A**) Raw data and images were obtained from patients diagnosed as PDAC in our hospital, cleaned, and pre-processed. Pancreatic CT images were processed through the AX-Unet pancreas segmentation model to obtain imaging features through mathematical formulas. (**B**) Clinical features and image features are shown in different colours, which were integrated as they became available. Data analysis was subsequently performed and associations between features were found. (**C**) AX-Unet model framework diagram [32]. The original image undergoes feature extraction and downsampling using an encoder. Subsequently, a modified ASPP module is employed to perform multi-scale feature extraction. Next, a decoder is utilized for upsampling and feature reconstruction. Finally, the output feature map is utilized for segmentation evaluation or loss calculation during model training.

**Figure 2 bioengineering-10-00828-f002:**
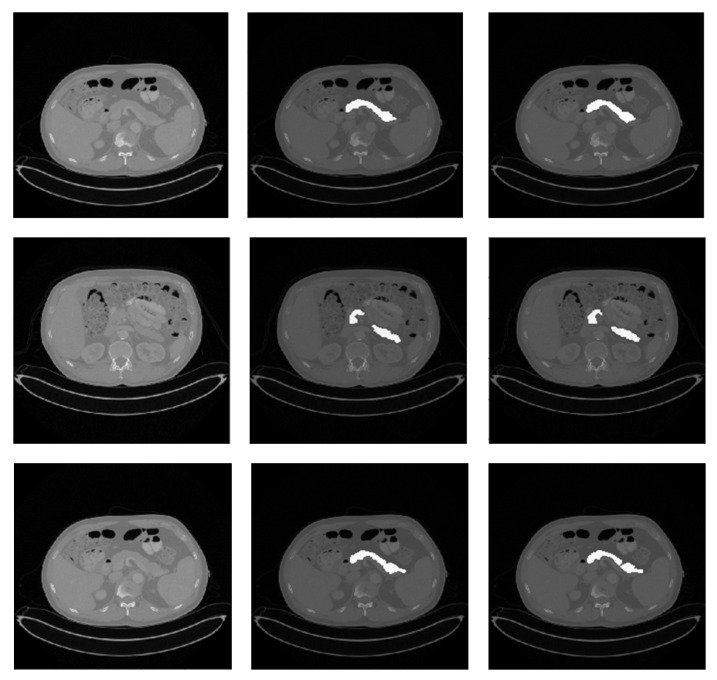
Qualitative pancreas segmentation results of AX-Unet model. The images in each row arranged from left to right depicts the original image, the ground-truth segmentation, and the segmentation generated by our AX-Unet model, respectively. The segmentation results are evidently of superior quality.

**Figure 3 bioengineering-10-00828-f003:**
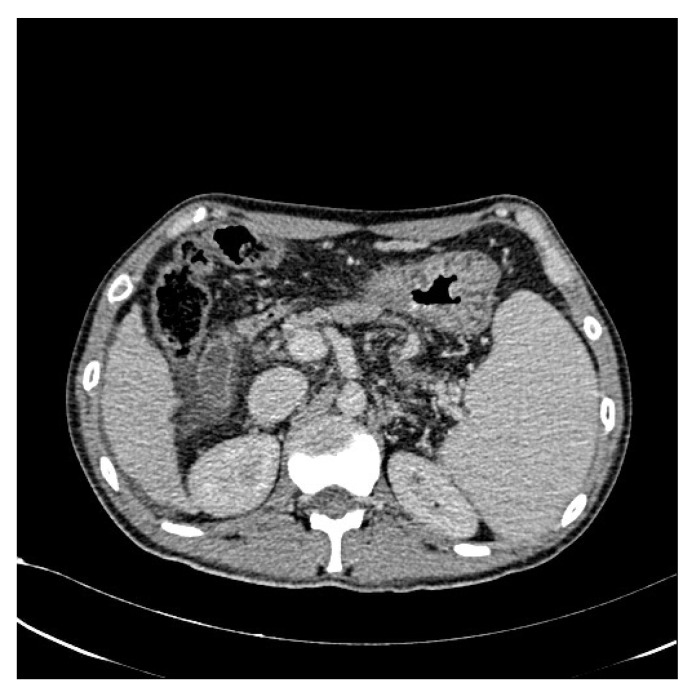
Original CT image.

**Figure 4 bioengineering-10-00828-f004:**
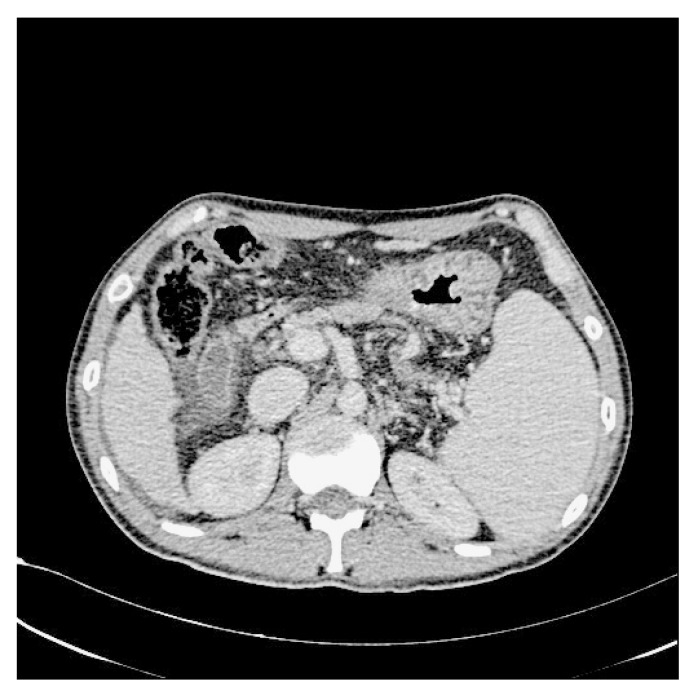
After increasing the contrast.

**Figure 5 bioengineering-10-00828-f005:**
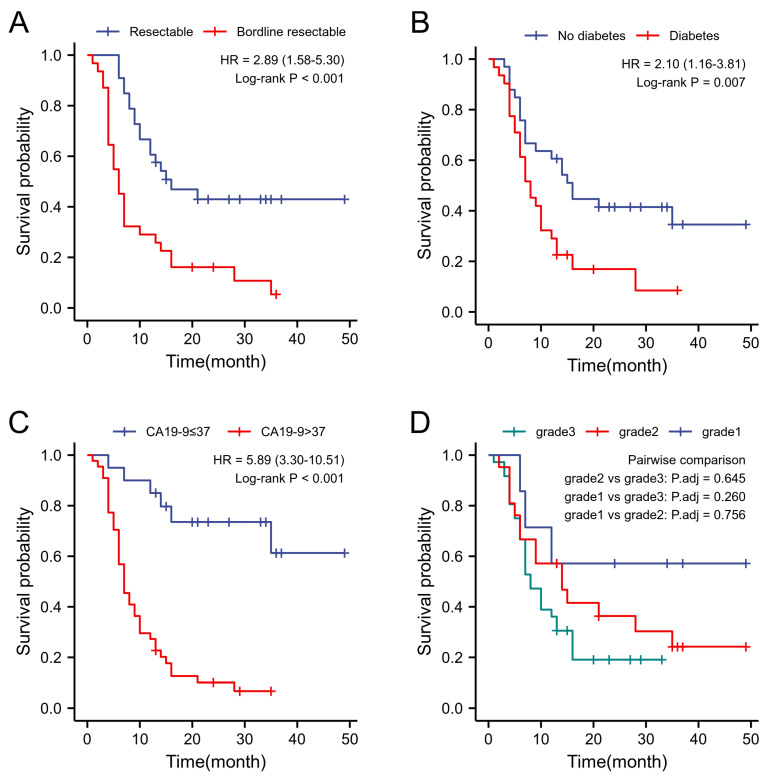
Kaplan–Meyer curves of RFS. (**A**) Survival differences between bordline resectable and resectable patients. (**B**) Survival differences between diabetics and non-diabetics. (**C**) Survival difference between patients with CA199 > 37 and those with ≤37. (**D**) Survival differences in patients with different pathological grades.

**Table 1 bioengineering-10-00828-t001:** AX-Unet performance results on our dataset, with the evaluation of our method described by the mean ± standard deviation.

Method	DSC (%)	Jaccard (%)	Recall (%)	Precision (%)
Our Dataset				
Unet-64	70.5 ± 3.8	59.3 ± 2.7	72.7 ± 2.5	70.8 ± 3.5
Unet-16	67.2 ± 2.6	54.8 ± 3.1	65.1 ± 1.7	69.3 ± 5.8
Bootom-up	70.8 ± 2.1	58.9 ± 1.8	73.3 ± 2.3	74.8 ± 2.5
Attention Unet	66.0 ± 3.2	52.6 ± 4.2	70.3 ± 2.1	71.5 ± 2.4
nn-Unet	80.7 ± 1.9	68.9 ± 3.5	83.3 ± 3.2	84.8 ± 4.7
**AX-Unet (Ours)**	**85.9 ± 3.5**	**74.2 ± 4.4**	**87.6 ± 2.4**	**89.7 ± 6.5**

**Table 2 bioengineering-10-00828-t002:** Baseline clinical characteristics of patients and survival analysis.

Characteristics	Univariate Analysis		Multivariate Analysis	
	mRFS	*p* Value	HR (95%Cl)	*p* Value
**Gender**		0.875		
Female (*n* = 23)	9 (1–37)			
Male (*n* = 41)	7 (3–49)			
**Age**		0.442		
≤65 (*n* = 48)	7 (1–49)			
>65 (*n* = 16)	13 (3–27)			
**BMI**		0.949		
≤24 (*n* = 44)	8 (2–49)			
>24 (*n* = 20)	10 (1–49)			
**History of DM ^1^**		0.007	1.16 (0.64∼2.24)	0.65
Yes (*n* = 31)	8 (1–36)			
No (*n* = 33)	15 (3–49)			
**Baseline CA19-9 ^2^** **(** μ **/mL** **)**		0.001	5.13 (2.05∼12.84)	0.01
≤37 (*n* = 20)	21 (4–49)			
>37 (*n* = 44)	6 (1–35)			
**Status of Resectability**		0.001	1.76 (0.90∼3.46)	0.1
RPC ^3^ (*n* = 33)	15 (3–49)			
BRPC ^4^ (*n* = 31)	7 (1–36)			
**N stage**		0.001	0.63 (0.32∼1.24)	0.18
N0 (*n* = 34)	15 (3–49)			
N1 (*n* = 30)	7 (1–36)			
**Histologic Grade**		0.124		
G1 (*n* = 7)	24 (6–49)			
G2 (*n* = 21)	14 (2–49)			
G3 (*n* = 36)	8 (1–33)			
**P53**		0.49		
≤50% (*n* = 45)	12 (2–49)			
>50% (*n* = 19)	13 (1–29)			
**Ki-67**		0.945		
≤50% (*n* = 47)	8 (2–49)			
>50% (*n* = 17)	14.77 ± 2.23			

^1^ DM: diabetes mellitus; ^2^ CA19-9: carbohydrate antigen 199; ^3^ RPC: resectable pancreatic cancer; ^4^ BRPC: borderline resectable pancreatic cancer.

**Table 3 bioengineering-10-00828-t003:** Association between significant texture features and clinical characteristics.

Characteristics	Significant Texture Features				
	Hu	Contrast	Entropy	Variance	Average
**Gender**	***p* = 0.32**	***p* = 0.008**	* **p** * **= 0.30**	***p* = 0.52**	* **p** * **= 0.01**
Female (*n* = 23)	42.57 (31.24–57.81)	6.24 (4.81–9.09)	8.46 (8.37–8.96)	15.47 (10.42–22.55)	0.40 (0.33–0.58)
Male (*n* = 41)	39.35 (29.59–62.95)	7.28 (4.16–12.69)	8.54 (8.35–8.95)	19.38 (9.36–26.69)	0.47 (0.30–0.68)
**Age**	***p* = 0.11**	***p* = 0.86**	***p* = 0.53**	***p* = 0.10**	***p* = 0.12**
≤65 (*n* = 48)	39.35 (29.59–62.95)	240.57 (205.67–269.61)	8.49 (8.35–8.96)	17.09 (9.37–22.68)	0.41 (0.30–0.67)
65 (*n* = 16)	48.62 (36.03–53.45)	7.10 (5.47–9.02)	8.57 (8.43–8.95)	19.93 (13.65–26.69)	0.52 (0.37–0.64)
**BMI**	***p* = 0.86**	***p* = 0.04**	***p* = 0.74**	***p* = 0.01**	***p* = 0.03**
≤24 (*n* = 44)	41.75 (29.59–60.37)	6.59 (4.81–9.02)	8.50 (8.35–8.96)	19.45 (10.42–26.69)	0.45 (0.33–0.68)
24 (*n* = 20)	39.43 (31.10–62.95)	7.52 (4.16–12.69)	8.56 (8.37–8.95)	14.58 (9.36–22.16)	0.40 (0.31–0.58)
**History of DM**	***p* = 0.01**	***p* = 0.83**	***p* = 0.11**	***p* = 0.01**	***p* = 0.35**
Yes (*n* = 31)	37.29 (29.59–53.18)	6.77 (4.16–12.69)	8.49 (8.35–8.95)	15.31 (9.36–22.16)	0.40 (0.30–0.63)
No (*n* = 33)	39.43 (31.10–62.95)	7.52 (4.16–12.69)	8.56 (8.37–8.95)	14.58 (9.36–22.16)	0.40 (0.31–0.58)
**Baseline** **CA19-9(μ/mL)**	***p* = 0.04**	***p* = 0.85**	***p* = 0.01**	***p* = 0.01**	***p* = 0.21**
≤37 (*n* = 20)	42.96 (39.35–62.95)	7.39 (4.16–9.65)	8.82 (8.57–8.95)	20.89 (12.01–26.69)	0.44 (0.40–0.64)
37 (*n* = 44)	40.21 (29.59–60.37)	6.68 (5.35–12.69)	8.48 (8.35–8.96)	14.81 (9.36–22.68)	0.41 (0.30–0.68)
**Status of** **Resectability**	***p* = 0.74**	***p* = 0.95**	***p* = 0.02**	***p* = 0.60**	***p* = 0.35**
RPC (*n* = 33)	40.21 (29.59–62.95)	6.60 (4.81–10.94)	8.56 (8.35–8.95)	19.45 (9.36–22.55)	0.41 (0.30–0.63)
BRPC (*n* = 31)	41.07 (31.25–60.37)	6.78 (4.16–12.69)	8.49 (8.37–8.96)	16.45 (10.42–26.69)	0.47 (0.33–0.68)
**N stage**	***p* = 0.96**	***p* = 0.14**	***p* = 0.02**	***p* = 0.16**	***p* = 0.79**
N0 (*n* = 34)	41.61 (31.10–62.95)	6.61 (4.81–9.65)	8.54 (8.37–8.96)	19.72 (11.04–26.69)	0.41 (0.30–0.64)
N1 (*n* = 30)	41.08 (29.59–60.37)	6.78 (4.16–12.69)	8.51 (8.36–8.95)	16.45 (9.37–22.68)	0.44 (0.33–0.67)
**Histologic** **grade**	***p* = 0.04**	***p* = 0.33**	***p* = 0.49**	***p* = 0.61**	***p* = 0.69**
G1 (*n* = 7)	54.73 (39.35–57.81)	8.47 (4.81–9.66)	8.57 (8.38–8.96)	14.45 (12.02–22.55)	0.41 (0.38–0.50)
G2 (*n* = 21)	42.96 (31.10–62.95)	6.68 (4.16–12.69)	8.53 (8.41–8.92)	18.73 (10.42–26.69)	0.43 (0.34–0.64)
G3 (*n* = 36)	38.69 (29.59–53.45)	6.78 (5.41–10.93)	8.49 (8.35–8.95)	18.22 (9.37–22.68)	0.44 (0.30–0.68)
**Recurrence**	***p* = 0.18**	***p* = 0.53**	***p* = 0.01**	***p* = 0.01**	***p* = 0.89**
Yes (*n* = 46)	41.08 (29.59–60.37)	6.78 (5.41–12.69)	8.49 (8.36–8.63)	15.31 (9.36–26.69)	0.43 (0.31–0.68)
No (*n* = 18)	45.35 (31.10–62.95)	6.60 (4.16–9.09)	8.88 (8.38–8.96)	20.74 (12.37–22.55)	0.44 (0.33–0.59)

**Table 4 bioengineering-10-00828-t004:** Kaplan–Meyer curves of RFS.

Characteristics	OR	95% Cl
**Gender**		
Male	9.45	1.01–88.71
Female	1	
**Status of Resectability**		
RPC	1	
BRPC	19.88	1.52–260.35
**Entropy**		
≤8.59	1	
>8.59	0.01	0.002–0.03

## Data Availability

The datasets generated during and/or analysed during the current study are available from the corresponding author upon reasonable request.

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
