# Peer review of "Predicting Recurrence in Pancreatic Ductal Adenocarcinoma after Radical Surgery Using an AX-Unet Pancreas Segmentation Model and Dynamic Nomogram"

_bioengineering, 2023, doi:10.3390/bioengineering10070828_

Round 1

Author Response

Thank you very much for taking the time to carefully review the article. Based on your suggestions, we have made detailed revisions to the article to make our expressions more perfect and the results more persuasive. The following pdf is our revision results.

Reviewer 2 Report

In the provided manuscript, the authors present a deep learning framework (AX-Unet) that while presented elsewhere [18] is now applied to pancreatic tissue classifications, which provide segmented regions from which predictive features can be extracted and used to make predictions on patient and disease characteristics.   Generally, it’s a good approach that I think the research community will appreciate. I have colleagues who are sure to be interested.   I have no major concerns, only some minor suggestions.   1. It’s mentioned that the AX-Unet is the result of combining something of three different frameworks (Unet, deepLabV, Xception), but I don’t think it’s explained what exactly was used from each (Unet probably is clear).   2. I think it would be worth considering whether figure 1 could be expanded or revised to some degree in order to clarify the workflow and high level approach used in the manuscript.  For example, texture features are not mentioned..  It’s maybe too high level and a little bit more description could be helpful to readers the first time through the paper.   3. It might be nice to show a comparison of image tiles that produce a very information rich / strongly classifying texture feature compared to an image tile that does not.  Or some visualization of the texture features —with their producing image— might be informative.   4. In the first section of the results, there’s some discussion about how the deep learning model structure has been modified, such as adding the ASPP module. I think it would be nice to have the model structure in it’s own figure with annotations that map to this text.  For example, what part of the DL model diagram is the ASPP module?   5. Great that you make the code available! Make sure the GitHub link works. Should be able to just copy it and paste it into a browser. Currently, I don’t think it does.   6. The GitHub documentation is not sufficient at this point. It would be important to provide some instruction to readers in order to be able to reproduce the results.   7. I believe that you will probably need to apply some multiple testing corrections to the statistical tests. You are repeatedly doing a Chi2 test, against the same clinical features, which requires some effort to be made towards multiple testing corrections.  In that case, you can report q-values, which will change the text very little. Or use Bayesian tests.  ;-)   8. While working on the statistics, perhaps you could include a notebook in the GitHub repo that uses the texture / other features and clinical features? That would be very nice.   9. Unfortunately, it’s hard to say increased entropy were associated with increased risk of recurrence, the OR is 0.01 and the 95% CI includes 0. Maybe that needs more evidence?   10. It’s stated in the discussion that the deconvolution module can decouple the channels which was confirmed by experiment, but I’m not sure the results of those experiments or clarity on this method (inputs, outputs, how?) are provided. Maybe I’m missing it somewhere.  

Author Response

(The authors gave the same response as above.)

Reviewer 3 Report

The work deals with the development of a pancreas segmentation model to analyze the recurrence of pancreatic ductal adenocarcinoma (PDAC). It is very interesting, original and well described. I have only some comments:

1. line 24: explicit the PDAC acronym the first time you use it.

2. line 63: replace (3) with 'Third' to be congruent with the previous listing mode.

3. in the Introduction section: Clarify which are the open problems to be solved and what contribution you want to make with this work. Furthermore it could be beneficial to adda a paragraph abouth the state of the art of PDAC segmentation, in order to compare your results with those already existing work (view also next comments 8-9).

4. line 86 and 310: use the acronym instead of expliciting "Pancreatic ductal adenocarcinoma".

5. Figure 1 B: some details are not visible, please increase resolution, especially for AX-Unet Model

6. Section 3.3.: I would move this section in the method section because it is about the patients characteristics.

7. Data listed in Table 3 could also be plotted in graphical format to facilitate the readibility.

8. In the results section please add a paragraph with comparison of the performace of the presented method with other already published work about PDAC segmentation.

9. In the Discussion section, please extend the discussion about limitations of the study and the comparison with other existing studies.

It is a very good work, only some modification are needed, from my point of view.

Author Response

(The authors gave the same response as above.)

Round 2

Reviewer 3 Report

I thank the authors for the consideration of my comments.

The revised version of the manuscript is now suitable for pubblication in my opinion.